# Evolution of Fetal Cardiac Imaging over the Last 20 Years

**DOI:** 10.3390/diagnostics13233509

**Published:** 2023-11-22

**Authors:** Nathalie Jeanne Magioli Bravo-Valenzuela, André Souza Malho, Caroline de Oliveira Nieblas, Pedro Teixeira Castro, Heron Werner, Edward Araujo Júnior

**Affiliations:** 1Department of Obstetrics, Paulista School of Medicine, Federal University of São Paulo (EPM-UNIFESP), São Paulo 05089-030, SP, Brazil; njmbravo@icloud.com (N.J.M.B.-V.); andremalho@hotmail.com (A.S.M.); 2Department of Pediatrics, Pediatric Cardiology, Federal University of Rio de Janeiro (UFRJ), Rio de Janeiro 21941-901, RJ, Brazil; 3Latin American Fetal Medicine Foundation (FMF-LA), Campinas 13025-070, SP, Brazil; 4Medical Course, Municipal University of São Caetano do Sul (USCS), São Caetano do Sul 09521-160, SP, Brazil; caroline.nieblas@uscsonline.com.br; 5Department of Obstetrics, University of Vassouras, Vassouras 27700-000, RJ, Brazil; pedrotcastro@gmail.com; 6Department of Fetal Medicine, Biodesign Laboratory DASA/PUC, Rio de Janeiro 21941-901, RJ, Brazil; heron.werner@gmail.com

**Keywords:** fetal heart, three-dimensional ultrasound, spatiotemporal image correlation, HDlive, fetal intelligent navigation echocardiography, artificial intelligence

## Abstract

The purpose of this article is to describe the evaluation of a variety of congenital heart diseases (CHDs) using three-dimensional (3D) ultrasound with different software, such as Cristal Vue, Realistic Vue, LumiFlow, and Spatiotemporal Image Correlation (STIC), with HDlive and HDlive Flow Silhouette modes. These technologies provide realistic images of the fetal heart and cardiac vessels using a fixed virtual light source that allows the operator to freely select a better light source position to enhance the cardiovascular anatomical details. In addition, Fetal Intelligent Navigation Echocardiography (FINE) technology, also known as “5D Heart” or “5D”, is a technology that enables the automatic reconstruction of the nine standard fetal echocardiographic views and can alert non-specialists to suspected CHD. Through the use of artificial intelligence, an ultrasound machine is able to perform automatic anatomical and functional measurements. In addition, hese technologies enable the reconstruction of fetal cardiac structures in realistic images, improving the depth perception and resolution of anatomic cardiac details and blood vessels compared to those of standard two-dimensional (2D) ultrasound.

## 1. Introduction

Congenital heart disease (CHD) is the most common birth defect and a leading cause of perinatal mortality related to birth defects [1]. The prenatal diagnosis of cardiac anomalies is a key element in improving the prognosis of structural CHD and optimizing the prevention of cardiovascular disease in hearts that have suffered a prenatal insult, such as fetuses with growth restriction. In this scenario, the development of prenatal cardiac ultrasound screening programs has improved the early detection of CHD. The detection rates have been improved by adding ventricular outflow tracts and superior mediastinal views to the classic standard four-chamber (4C) view during cardiac ultrasound [2].

The advent of advanced imaging technologies, such as three-dimensional (3D) ultrasound in the early 1990s and, in particular, Spatiotemporal Image Correlation (STIC) software in the early 2000s, enabled major technological advances in the 3D assessment of the fetal heart. STIC technology, as first described by De Vore et al. [3], involves the acquisition of approximately 150 two-dimensional images per second using a volumetric transducer during a single 7.5 to 15 s scan. The acquired 3D volumes contain a “block” of cardiac ultrasound images that allow the detailed analysis of cardiac anatomy (Figure 1A) [4]. This technology also provides measurements of the areas of the fetal heart valves and the volumes of the chambers and myocardial walls (Figure 1B). Some of the advantages of STIC in the assessment of the fetal heart include less operator dependence in obtaining ultrasound views of the heart, a shorter scan examination time, as the volume analysis can be performed in the absence of the pregnant woman (offline), and the ability to send the heart volumes via an Internet link for review by experts in tertiary centers (tele-STIC) [5,6].

In this scenario, new software called HDlive (General Electric Healthcare, Zipf, Austria) can be added to STIC to improve the quality of the ultrasound images. This technology differs from conventional 3D rendering methods in that the operator can freely select the light source at any angle, providing realistic sensations of the cardiac structures and their vessels, increasing the accuracy of cardiac anatomy detailing (Figure 2) [7,8,9,10].

Fetal Intelligent Navigation Echocardiography (FINE), also known as “5D Heart” or “5D”, presents nine standard fetal echocardiographic views by performing cardiac volume acquisition from a 4C chamber view of the fetal heart (Figure 3) [11]. Currently, with the advent of “artificial intelligence”, the addition of this technology to 5D Heart can provide alerts for possible CHD in the images obtained via automatic reconstructions of the cardiac ultrasound views. Artificial intelligence is a promising technological innovation that can alert non-specialists to the suspicion of CHD, provide automatic measurements of image acquisition, and reduce the differences between inter-observer measurements.

In this article, the authors describe a variety of advanced imaging technologies that, when available, are important tools to provide detailed, high-quality cardiac anatomy images. These advanced technologies may increase the prenatal detection of CHD, with a positive impact on reducing the perinatal mortality rate related to cardiac malformations.

## 2. Three-Dimensional Ultrasound with Spatiotemporal Image Correlation

Three-dimensional (3D) ultrasound with Spatiotemporal Image Correlation (STIC) was introduced in the early 2000s. This technology allows the volumetric acquisition of the fetal heart and its vessels, offering the ability to evaluate them in multiplanar and rendering modes, either static or in motion (four-dimensional—4D), reproducing a complete cardiac cycle [12,13].

The advantages of STIC are its higher sensitivity in the diagnosis of CHD, especially in conotruncal anomalies, and the shorter time of in-person ultrasound examination with the possible offline analysis of the fetal heart (after the pregnant woman has left the examination room). The details of morphology and function can be assessed using the rendering mode and, in complex cases, cardiac volumes can be sent via the Internet to reference centers specializing in fetal echocardiography. This technique is therefore suitable for ultrasound cardiac screening programs [14,15,16].

To obtain a cardiac volume using the STIC method, the examiner must obtain a 4C view of the fetal heart, whenever possible, with the fetal spine positioned as close to 6 o’clock as possible. Ideally, this image acquisition should be performed with the pregnant woman in apnea for a few seconds and the fetus at rest. The settings for this acquisition are: an opening angle between 20° and 40° and an acquisition time between 7.5 and 15 s according the gestational age [17]. By adjusting the brightness and color, STIC in the rendering mode allows the detailed evaluation of cardiac anatomy and measurement (area and volume) of a variety of structures of the fetal heart, such as the interventricular septum, valves, and papillary muscles [18,19,20,21,22]. STIC technology can reconstruct moving images and display them in grayscale, Doppler mode (inversion mode/color Doppler), B-flow imaging, and Tomographic Ultrasound Imaging (TUI) [23,24,25,26] (Figure 4, Figure 5 and Figure 6).

Grayscale STIC allows the assessment of the ventricular outflow tracts as well as the aortic and ductal arches. In this scenario, the B-flow imaging technique in STIC enhances the weak signals from blood and eliminates strong signals from nearby structures and is potentially superior to color Doppler in the assessment of large vessels and venous connections. B-flow imaging produces images similar to angiographic images (Figure 5 and Appendix A). The post-processing inversion mode (inverted flow) is another technique that analyzes fluid structures and inverts the grayscale voxels. The inversion mode allows the reconstruction of ventricles, aortic, and ductal arches, as well as abnormal venous connections and septal defects (Figure 6 and Appendix A).

Fetal cardiac volumes and functional parameters (stroke volume, cardiac output, and ejection fraction) can be assessed using STIC with the Virtual Organ Computer-Assisted Analysis (VOCAL) method. Each ventricular volume is obtained at the end of the diastole (maximum volume) and at the end of the systole (minimum volume), with the caliper positioned on the inner side of the basal and apical ventricular walls. After six sequential views, which are manually delineated for each ventricle, VOCAL provides the chamber volume and reconstructs its 3D image (Figure 7). Finally, the stroke volume, cardiac output, and ejection fraction of each ventricle are calculated from these measurements (Appendix A) [27,28,29,30].

## 3. HDlive

The new applications, HDlive, HDlive Flow, and HDlive Silhouette mode (General Electric Healthcare, Zipf, Austria), with STIC have the potential to provide a more realistic appearance of the cardiac structures and demonstrate blood flow with vitreous-like clarity. The HDlive Flow Silhouette mode has the ability to outline the walls of blood vessels, while showing the vessel lumen as semi-transparent (Figure 8). This technology demonstrates blood flow with a clarity similar to that of the vitreous (Appendix A). The accuracy of this technique can assist the team of professionals involved in the ultrasound assessment of the fetal heart by providing a better delineation of the blood vessels with a positive impact on the prenatal diagnosis of CHD [7,8,9,10,31,32]. HDlive Flow uses an examiner-adjustable light source to create light and shadow effects, increasing the depth perception of the 3D/4D color/power Doppler ultrasound. Hata et al. [33] demonstrated that the resolution and image quality of HDlive Flow are significantly higher than those obtained from conventional 3D/4D power Doppler, which may provide a better understanding of fetal cardiac anatomy.

Recently, Li et al. [34] demonstrated, in a retrospective study of 174 fetuses with conotruncal anomalies, that the accuracy of STIC with HDlive increases the detection of these cardiac anomalies, especially truncus arteriosus. The conotruncal anomalies detected in this study were described as follows: tetralogy of Fallot (58 cases), the transposition of the great arteries (30 cases), the double outlet of the right ventricle (26 cases), and 32 cases of truncus arteriosus. Complex CHD technology provides high-quality images that detail the anatomy, increasing the diagnostic accuracy [35].

## 4. Fetal Intelligent Navigation Echocardiography “5D Heart”

The Fetal Intelligent Navigation Echocardiography (FINE) technique, known as “5D Heart” (Samsung Healthcare, Gangwon, South Korea), consists of the automatic reconstruction of the nine standardized fetal echocardiographic landmarks after marking strategic cardiac anatomical points from the four-chamber view of the fetal heart. Using intelligent navigation, this software guides seven points to be marked by the examiner (the descending aorta, crux cordis, pulmonary valve, superior vena cava, and transverse aorta). The nine views of fetal echocardiography are automatically played in sequence, and the software itself labels the views (Appendix A). It is possible to zoom in on each view separately and to enhance the image separately using the brightness and contrast buttons. By applying FINE technology to STIC volume datasets, the examination of the fetal heart can be simplified with reduced operator dependency [11,36]. By adding color Doppler to FINE, the detection of CHD can be improved by assessing the Doppler flow characteristics of the heart vessels [37]. In a cross-sectional prospective study, Carrilho et al. [36] concluded that the quality of echocardiographic views obtained with the FINE technique was superior to that of the other techniques, such as the simple targeted arterial rendering (STAR) technique and the 4C view swing technique (FAST). 

## 5. Realistic Vue, Crystal Vue, and LumiFlow

Realistic Vue, Crystal Vue, and LumiFlow (Samsung Healthcare, Gangwon, South Korea) provide more detailed and realistic images of the fetus with instant processing, and Crystal Vue improves the visualization of internal and external structures in a single rendered image (Figure 9 and Figure 10 and Appendix A). LumiFlow is a post-processing and shading technique used to better assess the fetal micro- and macrovasculature of the fetus. This technology can be added to existing Doppler images, such as color/power, to simulate a 3D pace view of the fetal vasculature. With LumiFlow, the center of the vessel is displayed in a slightly lighter color due to its faster velocity, with a darker border around the vessels where blood velocities are slower (Figure 11) [38,39].

## 6. Artificial Intelligence

Congenital heart disease (CHD) can present as a wide variety of cardiac malformations, especially the more complex ones, requiring the team involved in the diagnosis to be well versed in the diversity of cardiac defects. As a result, many CHDs remain undiagnosed during ultrasound. Considering that CHD is a major cause of infant mortality and that prenatal diagnosis is crucial in this regard, the impact of this undiagnosed condition is unfavorable for the survival of these patients. In this scenario, artificial intelligence (AI) can assist non-specialists in diagnosing CHD. Machine learning (ML), which can be defined as the use of computer programs that improve over time, is an integral part of this technology. In addition, Deep Learning (DL) is a specific type of ML that uses neural networks organized into many layers, allowing the interpretation of the data provided, such as image classification in computer vision. As an example, DL is able to automatically detect the prognosis of CHD once the diagnosis is known [40,41].

With ML, the ultrasound machine is able to automatically obtain cardiac measurements based on the identification of anatomical cardiac structures, reducing the scan time and inter-observer measurement variation. In this scenario, not only anatomical, but also functional parameters (such as the myocardial performance index) can be automatically measured. With the advances in AI, this technology has been increasingly added to ultrasound devices, such as obstetric ultrasound, and is applicable to three areas: structure identification, automatic measurements, and the classification of diagnosis. AI software (Heart Assist and MPI+—Samsung Healthcare, Gangwon, South Korea) can identify the fetal heart structures and perform automatic biometric and the right and left myocardial performance index measurements, respectively (Figure 12 and Figure 13) [42,43,44].

## 7. Isomerism

The examination of the upper abdomen of the fetus via ultrasound provides a situs assessment. In the transverse abdominal view, if the situs is normal (*situs solitus*), the aorta is a left-sided vessel and is located posterior to the inferior vena cava. Left atrial isomerism is a condition in which there are two left-sided organs, and right atrial isomerism is the opposite situation (for example, the atria and lungs have the same shape). In left atrial isomerism, the venous vessel is posterior to the aorta due to the absence of the hepatic segment of the inferior vena cava, with drainage via the azygos or hemiazygos veins. The aorta and the azygos or hemiazygos veins are juxtaposed (a “double-vessel” sign) [45,46]. In general, abnormalities are associated with more complex CHDs, such as an unbalanced atrioventricular septal defect [47]. Conversely, in right atrial isomerism, there is a double right-sided atrium. Therefore, in the upper transverse abdominal view, the two vessels (venous and arterial) are located on the right side of the fetal spine. In this scenario, advanced technologies (5D Heart, HDlive, and HDlive Silhouette modes) are useful tools for the detailed anatomical diagnosis of these complex anomalies, as they reduce the ultrasound scan time and provide high-quality images (Figure 14 and Figure 15, Appendix A).

## 8. Atrioventricular Septal Defect

An atrioventricular septal defect (AVSD) is a CHD that results from the failure of the atrioventricular septum to develop, which can lead to abnormalities of the atrioventricular valves, an ostium primum atrial septal defect (ASD), and an inlet ventricular septal defect (VSD). The complete form of an AVSD is characterized by a common atrioventricular valve, a ostium primum-type ASD, and a large inlet ventricular septal defect (VSD). In other forms of AVSDs, the right and left atrioventricular valves are present with a small VSD or intact interventricular septum. The complete form of an AVSD is strongly associated with syndromes such as trisomy 21. Unbalanced complete AVSD (=marked discrepancy between right and left ventricle) is typically associated with heterotaxy syndromes such as left isomerism. The complete form of an AVSD, can be identified on the 4C view of the fetal heart with color Doppler by an “H-shaped” sign due to the absence of the atrioventricular septum. An AVSD may be associated with other cardiac defects such as tetralogy of Fallot, double outlet right ventricle, and anomalies with outflow tract obstruction. Advanced technologies can provide high-quality images to facilitate early detection during the first trimester cardiac screening in cases with increased nuchal translucency and in complex cases, such as it is when associated with other cardiac anomalies (Figure 16 and Figure 17) [2,3,6,48].

## 9. Tetralogy of Fallot

Tetralogy of Fallot is the most common conotruncal anomaly and even the most common cyanogenic CHD. Classically, it is characterized by a tetralogy of morphological features, as follows: the obstruction of the right ventricular outflow tract (subpulmonary obstruction), a VSD, the rightward deviation of the aorta (overriding = biventricular connection of the aorta), and right ventricular hypertrophy. In general, right ventricle hypertrophy is not visible in the fetal period, especially with two-dimensional (2D) ultrasound (Figure 18) [49]. Right-sided aortic arch (from 13 to 25% of tetralogy of Fallot cases), muscular VSDs, and AVSDs are some of the cardiac anomalies most commonly associated with tetralogy of Fallot. The extra-cardiac defects associated with it are the following: trisomy (13, 18, and 21), CHARGE, VATER, pentalogy of Cantrell, omphalocele, and Di George syndrome (deletion 22q11.2); in the latter one, the evaluation of the thymus is important [50].

For the diagnosis of tetralogy of Fallot, the outflow tract views are critical, and advanced imaging techniques will improve the accuracy and timing of the ultrasound examination, as in classic tetralogy of Fallot, the image of the fetal heart in the 4C view is normal [34,51]. Tetralogy of Fallot with subpulmonary stenosis is the classic form of this CHD, while the extreme form is pulmonary atresia. There is also a rare form of tetralogy of Fallot in which the pulmonary valve is absent [52,53].

The standard 4C view using 2D ultrasound is generally normal in the classic forms of tetralogy of Fallot, where the evaluation of the outflow tract views is critical. Advanced technologies such as 5D Heart can help to reduce the time of the scan examination by allowing the automatic reconstruction of the nine echocardiographic views. Therefore, in the classical forms of tetralogy of Fallot, this diagnosis becomes easier by drawing attention to the small pulmonary artery in the three vessels and trachea view and the overring of the aorta in the left ventricle outflow tract view [51,52]. In extreme cases, the presence of retrograde flow from the ductus arteriosus to the pulmonary artery confirms the atresia, which can be easily detected with advanced technologies (Figure 19). Conversely, in the forms of tetralogy of Fallot in which the pulmonary valve is absent, the right and left pulmonary arteries are enlarged (Figure 20) [53].

In fact, the advanced technologies reveal the anatomical details due to high-quality imaging, such as in non-classical forms of tetralogy of Fallot or when it is associated with other heart defects such as an AVSD (Figure 21).

## 10. Transposition of the Great Arteries

The complete transposition of the great arteries (TGA) is characterized by concordant atrioventricular connection with discordant ventriculoarterial connection, and therefore, does not include the hearts with atrial isomerism. The interventricular septum does not have the usual curvature of the normal heart, reflecting the outflow tracts of the ventricles with the great arteries “in parallel”. In simple TGA, the image of the fetal heart in the 4C view is normal, with altered views of the ventricular outflow tracts due to ventriculo-arterial concordance and ventriculoarterial discordance with the great arteries in a parallel relationship. Double outlet right ventricular outflow tract with the anterior aorta, known as a Taussig–Bing anomaly, remains a challenging differential diagnosis for TGA. The presence of a subpulmonary ventricular septal defect with both great arteries originating predominantly (>50%) from the morphologically right ventricle and subpulmonary stenosis may be helpful in the diagnosis of a Taussig–Bing anomaly [54]. Therefore, its prenatal detection rate remains low. In these cases, advanced technologies can be important diagnostic tools by providing anatomical details of the ventricular outflow tracts (Figure 22, Figure 23 and Figure 24) [34,55,56]. It is important to note that 3D images can sometimes be more difficult to obtain and interpret than standard 2D images for an examiner who is not experienced in advanced ultrasound technologies and who is not an expert in cardiac anatomy.

## 11. Anomalous Venous Return

The identification of anomalous pulmonary vein connections can be difficult in the fetus, especially in the partial forms. Considering that the total forms of anomalous drainage of the pulmonary veins are a critical CHD, prenatal diagnosis is crucial to improve the prognosis of these patients. In this setting, it is essential to identify the presence of one or more pulmonary veins draining into the left atrium during routine cardiac ultrasound. To rule out this diagnosis, it is possible to identify the presence of left and right pulmonary veins draining into the left atrium in the 4C view with the aid of color Doppler. In total anomalous pulmonary venous return (TAPVR), all the pulmonary veins drain directly into the morphologically right atrium or through a systemic vein. In these cases, the imaging techniques are very important diagnostic tools for the identification and anatomical detailing of the anomalous venous return. Based on the level of the anomalous connection, TAPVR is generally classified into four major types: supracardiac (Type I), cardiac (Type II), infracardiac (Type III) (Figure 25), and mixed levels of connections (Type IV) [57].

## 12. Right Aortic Arch

When the aortic arch is in an abnormal position, such as a right arch, the left subclavian artery has a retroesophageal/retrotracheal course that may cause the compression of the esophagus and/or trachea (vascular ring). A U-shaped image allows the diagnosis of this anomaly in the three-vessels view (Figure 26). The double arch of the aorta is the only form of vascular ring in which the vascular ring consists entirely of vessels. In a double arch, the right arch is usually larger than the left arch. The two aortic arches, the pulmonary trunk, and the ductus arteriosus form the vascular ring in the shape of a trident [58].

## 13. Conclusions

In conclusion, advanced imaging technologies, when available, may increase the prenatal detection of CHD by improving the depth perception and resolution of anatomic cardiac details and blood vessels compared with those of standard 2D ultrasound. Three-dimensional techniques, such as STIC and FINE, allow the reconstruction (real-time or offline) of fetal cardiac structures in realistic views based on a basic cardiac ultrasound view, such as the 4C view. With the use of FINE and artificial intelligence, automatic cardiac ultrasound views and automatic measurements of the anatomical and functional parameters of the fetal heart are provided, highlighting abnormalities. As a result, the examination scan time can be reduced, with a better understanding of the fetal heart and vessels, especially when the advanced US technologies enable the automatic reconstruction of echocardiographic views and artificial intelligence.

## Figures and Tables

**Figure 1 diagnostics-13-03509-f001:**
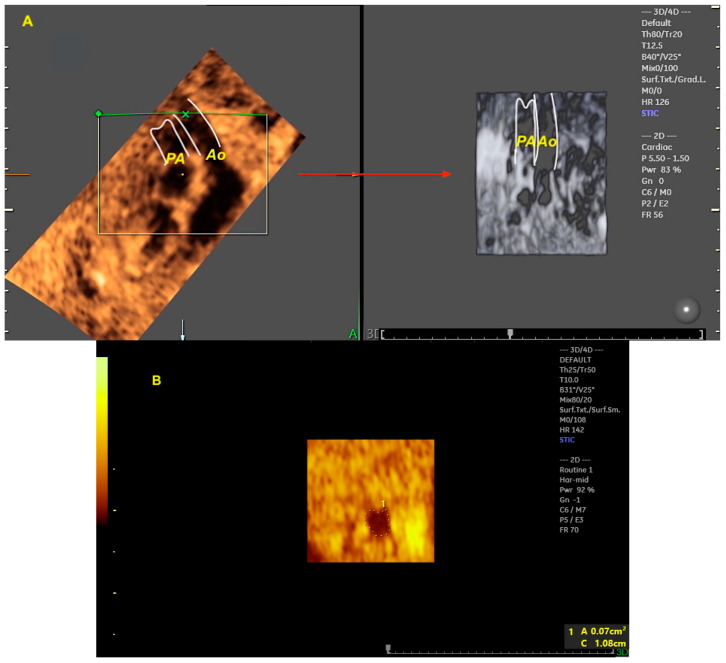
Spatial Temporal Imaging Correlation (STIC) rendering mode. STIC navigation from a 4-chamber view acquisition in a fetus with the transposition of the great arteries, allowing the reconstruction of the ventricular outflow tracts. In order to render the ventricular outflow vessels, the region of interest (ROI, green line) is adequately positioned enabling the image of the aorta and pulmonary arteries with sepia color mapping and in the adjacent image (red arrows) with inversion mode postprocessing (**A**). Note the parallel relationship of the aorta and pulmonary arteries. (1) Manual measurement of the ascending aortic area (0.07 cm^2^) in the STIC rendering mode (**B**). 1: ascending aorta area; Ao: aorta; PA: pulmonary artery.

**Figure 2 diagnostics-13-03509-f002:**
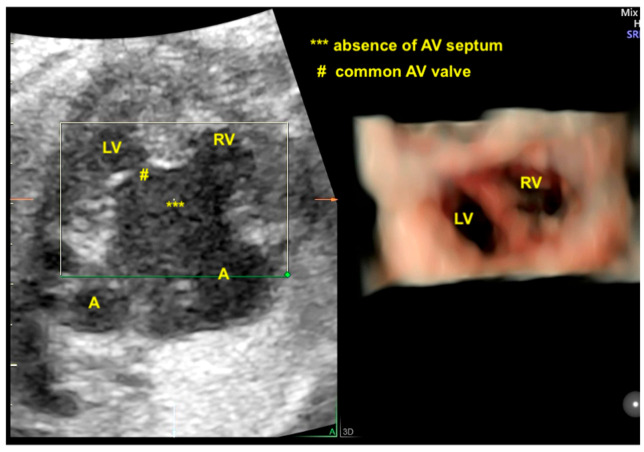
Spatiotemporal Image Correlation with HDlive navigation from a 4-chamber view, providing realistic imaging of the discrepancy between the ventricles in a case of unbalanced atrioventricular septal defect. Note the hypoplasia of the LV. LV: left ventricle; LA: left atrium; A: atrium; RV: right ventricle; A: right or left atrium; AV: atrioventricular.

**Figure 3 diagnostics-13-03509-f003:**
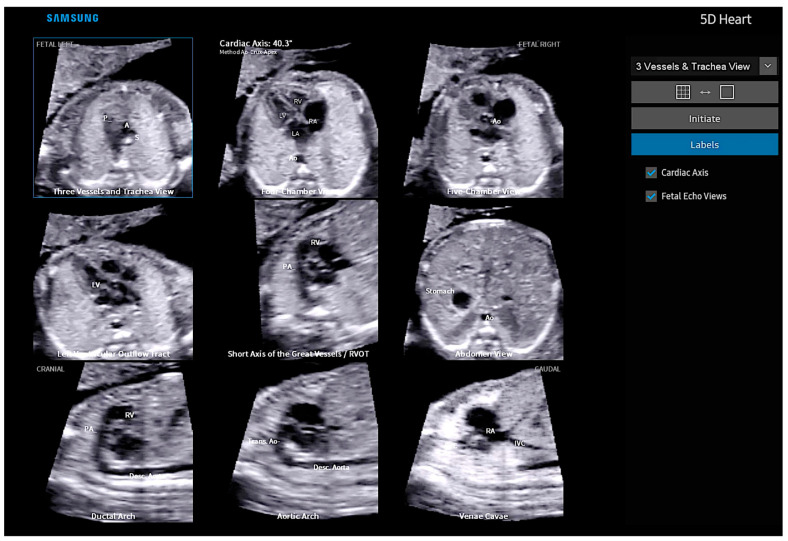
Fetal Intelligent Navigation Echocardiography (FINE), also known as “5D Heart”, on a normal heart. Note the automatic reconstruction of the 9 standard views of fetal echocardiography from a 4-chamber view. LV: left ventricle; LA: left atrium; RA: right atrium; RV: right ventricle; Ao: aorta; PA: pulmonary artery; SVC: superior vena cava; IVC: inferior vena cava; RVOT: right ventricle outflow tract.

**Figure 4 diagnostics-13-03509-f004:**
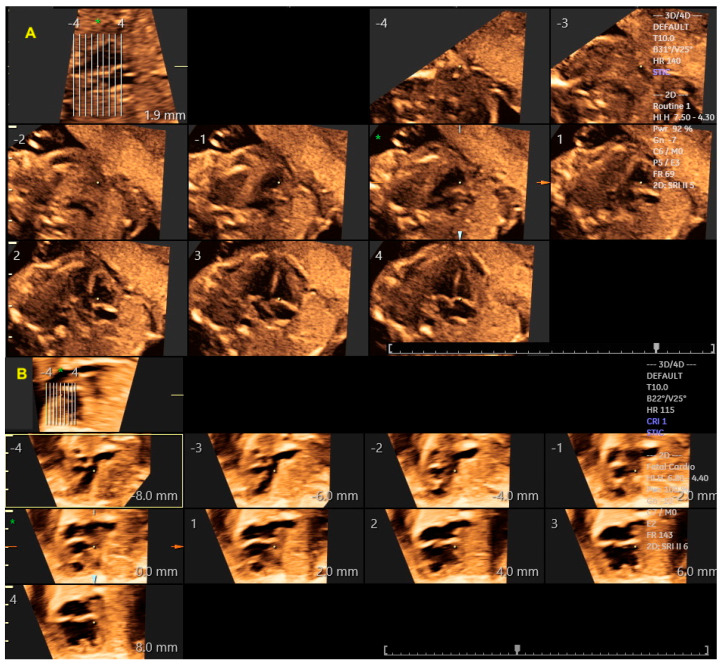
Spatiotemporal Image Correlation with Tomographic Ultrasound Imaging (TUI) technique, providing automatic slices with visualization of the sequential axial views of a normal heart (**A**) and in the case of atrioventricular septal defect with double right ventricle outflow tract (**B**). Interslice distance 4 mm; * 0 mm. Asterisk and arrows: Automatic slices 4 mm * point 0.

**Figure 5 diagnostics-13-03509-f005:**
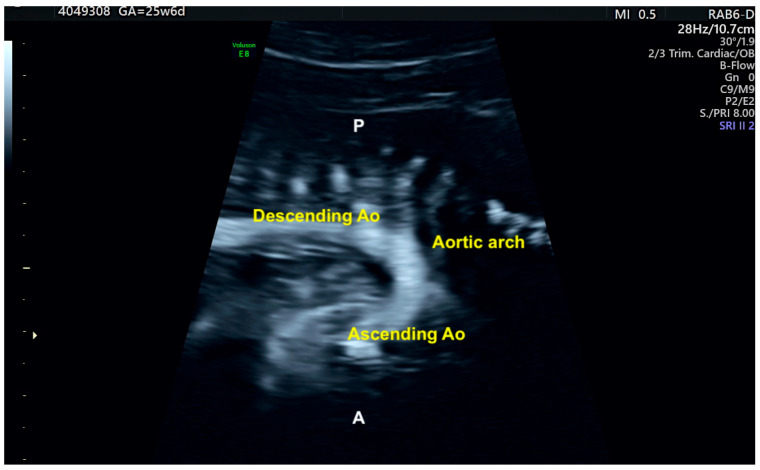
Aortic arch imaging using B-flow Imaging with Spatiotemporal Image Correlation method on fetus with normal heart. This technique encodes the blood signal and provides high-quality images similar to aortic angiography. Ao: aorta; P: posterior; A: anterior.

**Figure 6 diagnostics-13-03509-f006:**
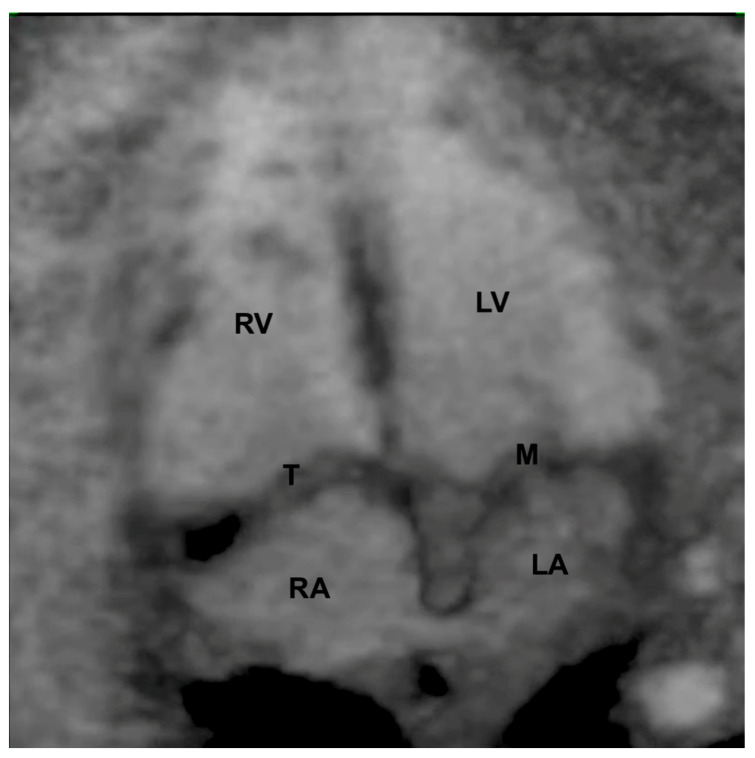
Inversion mode postprocessing in the Spatiotemporal Image Correlation method in a 4-chamber view of a normal heart. LV: left ventricle; LA: left atrium; RA: right atrium; RV: right ventricle; T: tricuspid valve; M: mitral valve.

**Figure 7 diagnostics-13-03509-f007:**
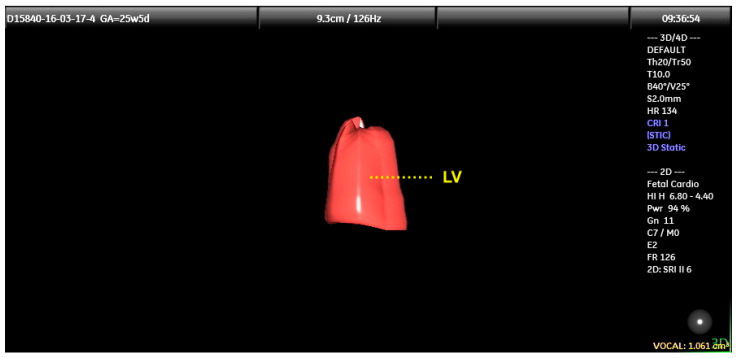
Reconstructed 3D image of left ventricular (LV) diastolic volume (LVD) using Spatiotemporal Image Correlation with Virtual Organ Computer-aided Analysis (VOCAL) on a fetus from a pregestational diabetic woman at 25 + 5 weeks gestation. The LVD volume was obtained from the end diastole (maximum diameter of that ventricle). To perform this measurement, the calipers were positioned at the inner edge of the mitral valve and at the endocardial edge of the ventricle. After six consecutive planes were obtained, the VOCAL provided the LV diastolic volume (1.061 cm^3^) and reconstructed the 3D image. Heart rate (HR) = 124 bpm. LV: left ventricle.

**Figure 8 diagnostics-13-03509-f008:**
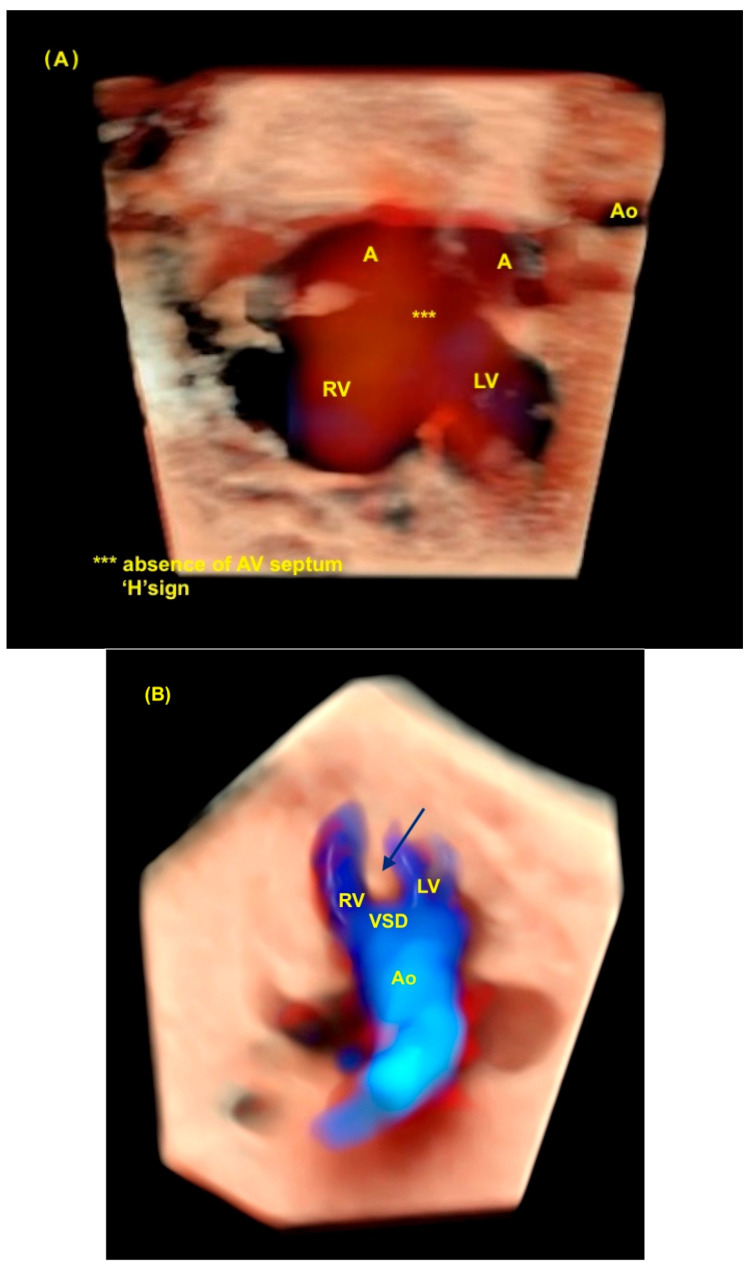
Spatiotemporal Image Correlation with HDlive Silhouette and HDlive Flow techniques of fetus with atrioventricular septal defect and tetralogy of Fallot. (**A**) Note the “H” sign (***) during diastole in the four-chamber view of the heart. The “H” sign is an important clue to the ultrasound diagnosis of total atrioventricular septal defect. (**B**) Note the superposition of the aorta in the ventricular outflow tract view due to the dextroposition of the aorta, which is an important clue to suspect the diagnosis of tetralogy of Fallot (interventricular septum—blue arrow). LV: left ventricle; A: atrium (left and right); RV: right ventricle; AV: atrioventricular; Ao: aorta; VSD: ventricular septal defect.

**Figure 9 diagnostics-13-03509-f009:**
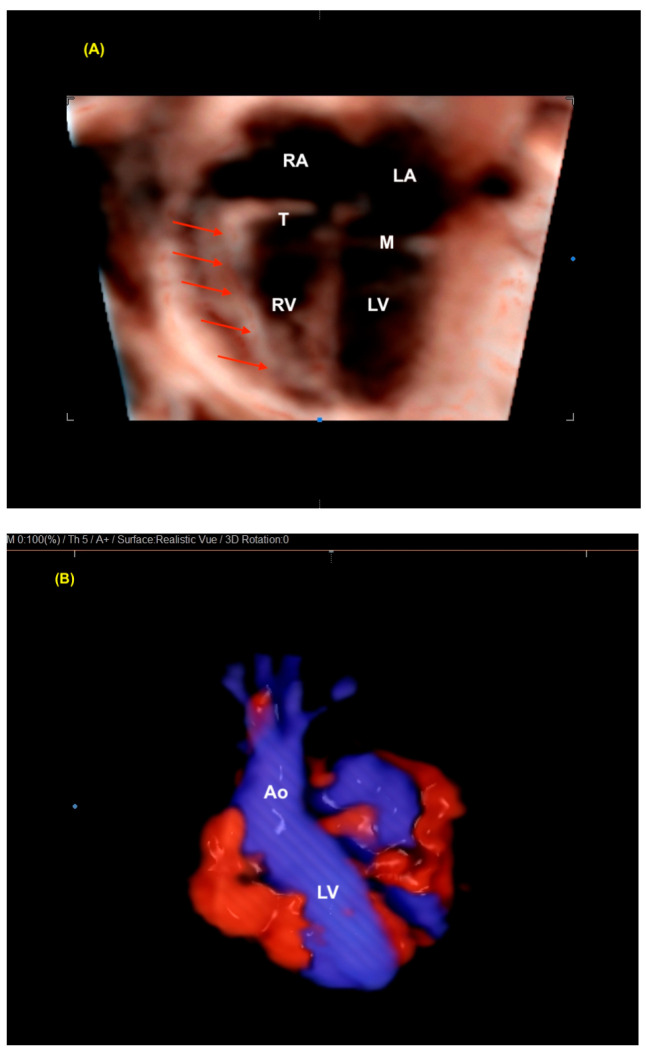
(**A**) Realistic Vue surface of a 4-chamber view in a case of tetralogy of Fallot (red arrows: RV hypertrophy). The hypertrophy of the RV is easily seen in this fetus with tetralogy of Fallot because this technology allows the detailed analysis of the cardiac anatomy. (**B**) Realistic color of the LV outflow tract in a fetus with a normal heart. Note the realistic, high-quality image. LV: left ventricle; RV: right ventricle; RA: right atrium; LA: left atrium; M: mitral valve; T: tricuspid valve; Ao: aorta.

**Figure 10 diagnostics-13-03509-f010:**
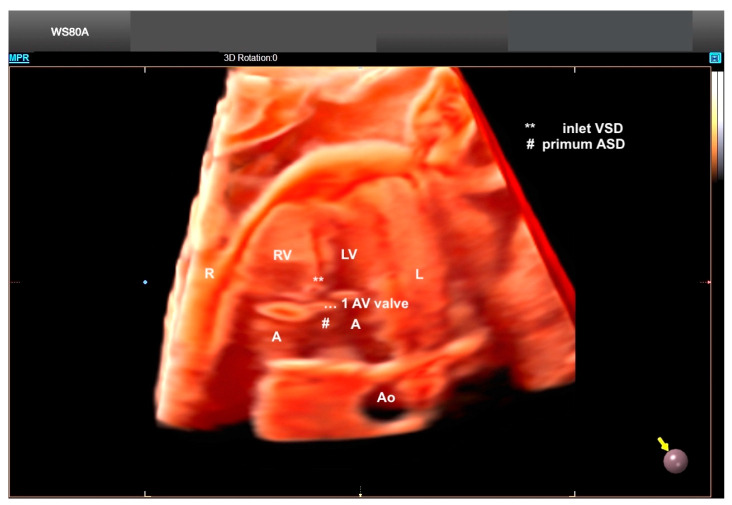
Crystal Vue and Realistic Vue in a case of unbalanced AV septal defect and left atrial isomerism. Note the small LV. Virtual light source position, 10 o’clock. Rotation, 0°. LV: left ventricle; A: atrium (left-sided morphological atrium); RV: right ventricle; AV: atrioventricular; Ao: aorta; R: right side of the fetus; L: left side of the fetus; VSD: ventricular septal defect; ASD: atrial septal defect.

**Figure 11 diagnostics-13-03509-f011:**
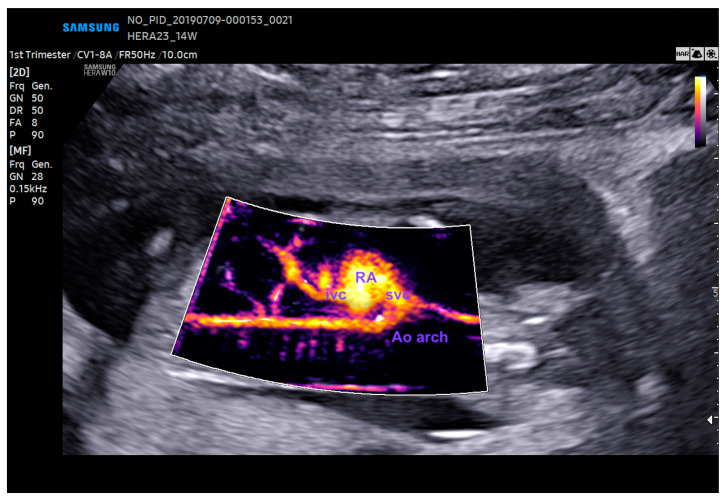
The aortic arch as assessed by LumiFlow at 14 weeks gestation. RA: right atrium; Ao: aorta; svc: superior vena cava; ivc: inferior vena cava.

**Figure 12 diagnostics-13-03509-f012:**
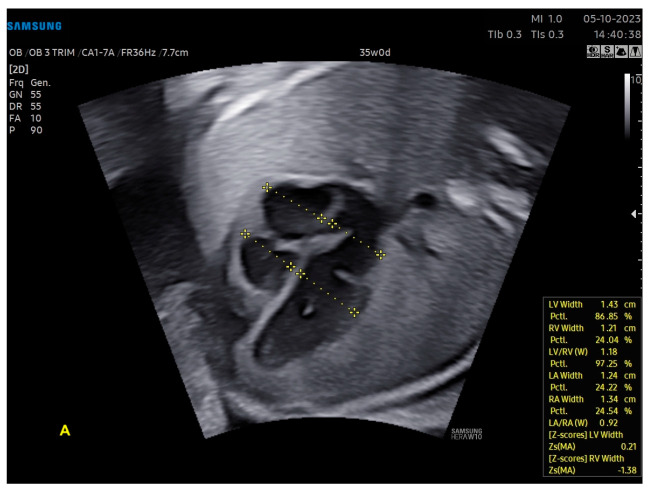
Artificial Intelligence (Heart Assistance). The cardiac ultrasound device is able to recognize anatomical heart structures and perform automatic measurements: (**A**) ventricles (width of atria and ventricles); (**B**) left ventricular outflow tract (atrioventricular annulus and ascending aorta); (**C**) aorta in sagittal view (ascending, transverse, isthmus, and descending aortas). Note that the images include the z-scores of these structures, which were also calculated automatically. LV: left ventricle; LA: left atrium; RA: right atrium; RV: right ventricle; Asc. aorta: ascending aorta; Desc. aorta: descending aorta; Transvers. aorta: transverse aorta.

**Figure 13 diagnostics-13-03509-f013:**
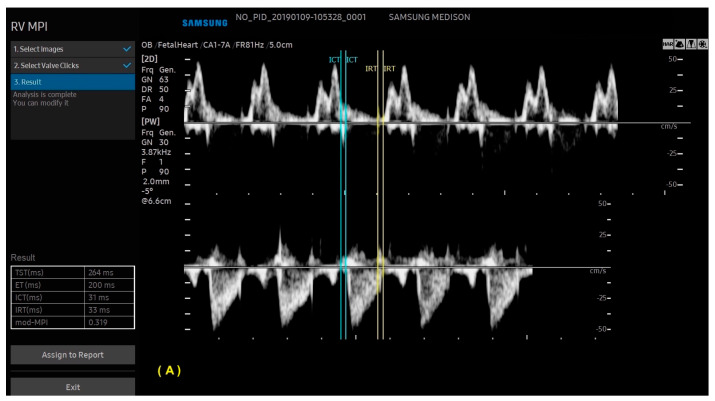
Artificial Intelligence (MPI+) The ultrasound machine is able to select Doppler images of the ventricular inflow and outflow tracts, identify valve clicks, and perform cardiac cycle time measurements. (**A**) Modified myocardial performance index (mod-MPI) was automatically calculated for the right (**A**) and left (**B**) ventricles of a fetus in the first trimester of pregnancy. LV: left ventricle; RV: right ventricle; ET: ejection time; ICT: isovolumetric contraction time; IRT: isovolumetric relaxation time.

**Figure 14 diagnostics-13-03509-f014:**
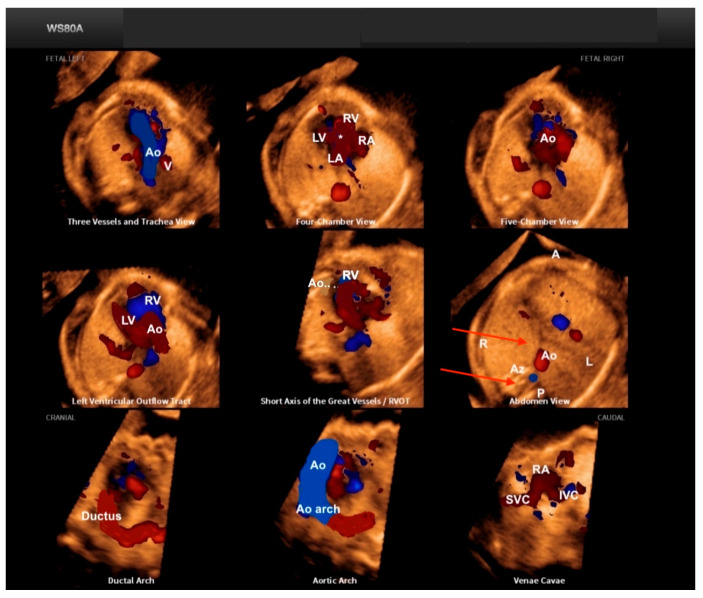
Fetal Intelligent Navigation Echocardiography (FINE), also known as “5D Heart”, in a case of left atrial isomerism, with a complete atrioventricular septal defect of the anterior aorta. Note: the “double vessel sign” (red arrows: aorta and azygos side by side in the upper abdominal view; the venous vessel is posterior) and (*) the “H” sign in the four-chamber view. Note: the aorta arises from the RV in the RV outflow tract view and the presence of two vessels (aorta and superior vena cava) instead of three vessels in the three vessels and trachea view. LV, left ventricle; LA, left atrium; RA, right atrium; RV, right ventricle; Ao, aorta; V, vein (superior vena cava in the three vessels and trachea view); Az, azygous; P, posterior; A, anterior; L, left side; R, right side; SVC, superior vena cava; IVC, inferior vena cava; RVOT, right ventricle outflow tract.

**Figure 15 diagnostics-13-03509-f015:**
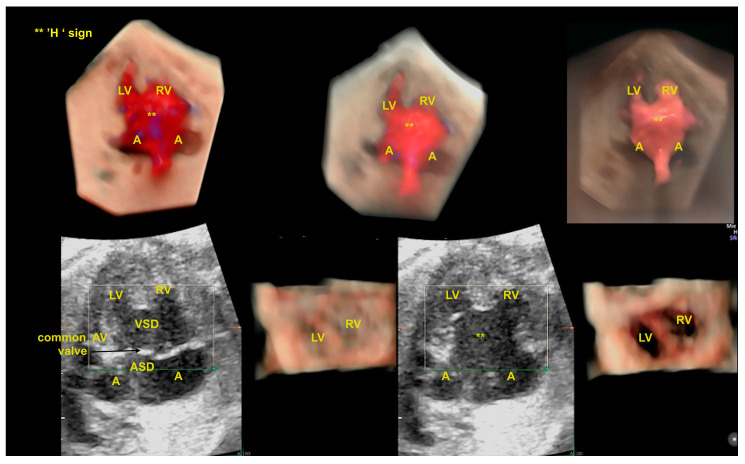
Spatiotemporal Image Correlation with HDlive Silhouette and HDlive Flow techniques in a fetus with left isomerism and a complete atrioventricular septal defect (AVSD). Completely unbalanced AVSD can be recognized in an abnormal 4-chamber view: note the common atrioventricular valve (black arrow), the absence of the AV septum (** ‘H’ sign), and the unequal-sized ventricles. The region of interest (ROI, green line) is adequately positioned enabling the detailed anatomy. A: morphologically left atrium; LV: left ventricle; RV: right ventricle; AV: atrioventricular; ASD: atrial septal defect; VSD: ventricular septal defect.

**Figure 16 diagnostics-13-03509-f016:**
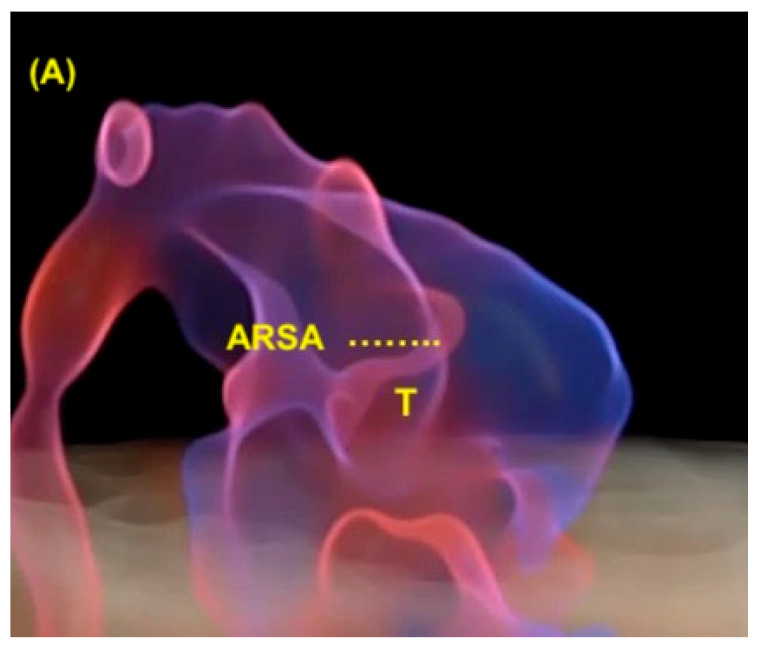
Note the realistic cardiac images using Spatiotemporal Image Correlation with HDlive Flow technique in a fetus with trisomy 21, left atrial isomerism, complete atrioventricular septal defect, and anomalous right subclavian artery (ARSA). (**A**) Three vessels and trachea view showing the ARSA with retrotracheal course. (**B**) Sagittal abdominal view showing the aorta and the hemiazygos (left atrial isomerism). Ao; aorta; Hz: hemiazygos; PA; pulmonary artery; ARSA: anomalous right subclavian artery; T: trachea (^……..^) A: anterior; P: posterior.

**Figure 17 diagnostics-13-03509-f017:**
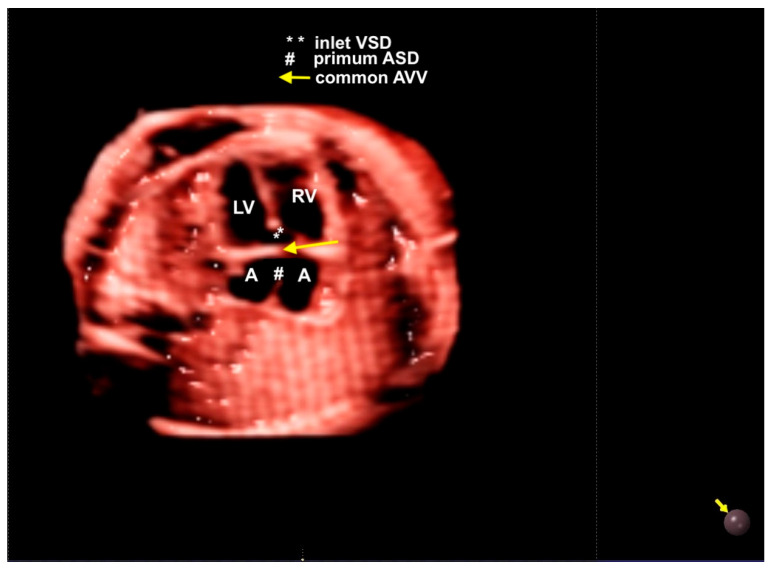
Realistic Vue with Natural Vue and Transparency mode in a case of complete atrioventricular septal defect. Note the high quality of the image. Virtual light source position; 10 o’clock. LV: left ventricle; LA: left atrium; LV: left heart; RV: right ventricle; VSD: ventricular septal defect; ASD: atrial septal defect; AVV: atrioventricular valve.

**Figure 18 diagnostics-13-03509-f018:**
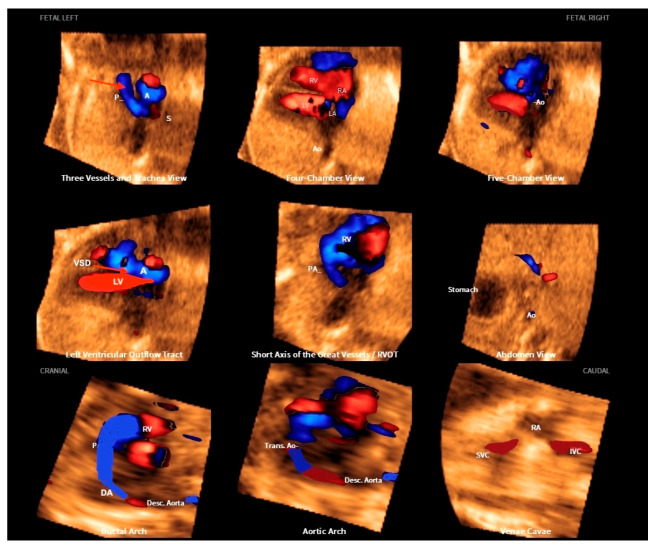
Fetal Intelligent Navigation Echocardiography (FINE), also known as “5D Heart”, in a case of tetralogy of Fallot. Note the small pulmonary artery in the three vessels and trachea view and the overriding of the aorta in the left ventricle outflow tract view (red arrows). A or Ao, aorta; P, pulmonary artery; LV, left ventricle; LA, left atrium; LV, left ventricle; RV, right ventricle; VSD, ventricular septal defect; DA, ductus arteriosus; SVC or S, superior vena cava; IVC, inferior vena cava; Trans. Ao, transverse aorta; Desc. Aorta, descending aorta; RVOT, right ventricle outflow tract.

**Figure 19 diagnostics-13-03509-f019:**
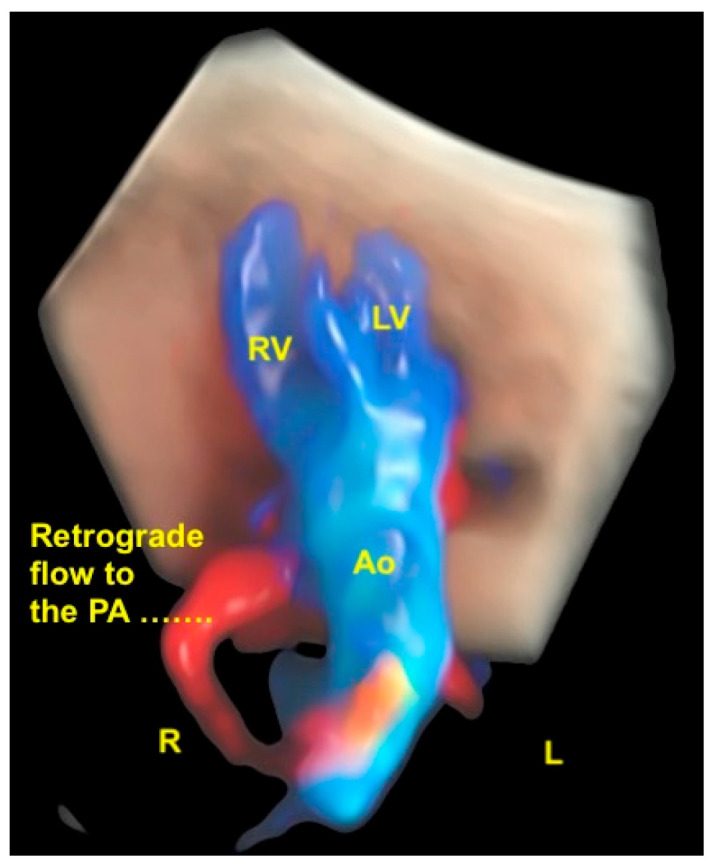
Tetralogy of Fallot with pulmonary atresia using Spatiotemporal Image Correlation with HDlive Flow Silhouette technique. Note the overriding of the aorta and the retrograde flow from the ductus arteriosus to the pulmonary artery (= pulmonary atresia). L, left side; R, right side; RV, right ventricle; LV, left ventricle; Ao, aorta; PA, pulmonary artery.

**Figure 20 diagnostics-13-03509-f020:**
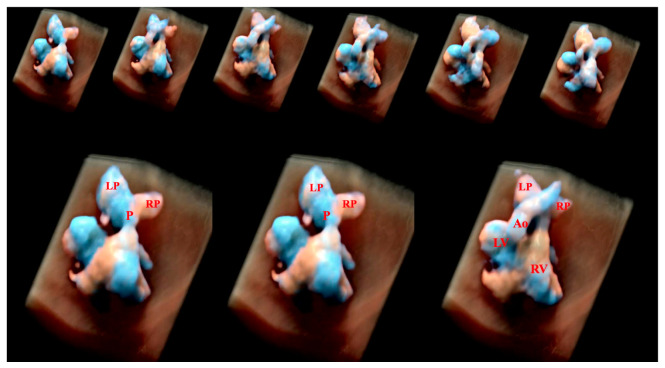
Tetralogy of Fallot with absent pulmonary valve using Spatiotemporal Image Correlation with HDlive Flow Silhouette technique. Note the dilation of the right and left pulmonary arteries. P: main pulmonary artery; LP: left pulmonary artery; RP: right pulmonary artery; Ao: aorta; LV: left ventricle; RV: right ventricle.

**Figure 21 diagnostics-13-03509-f021:**
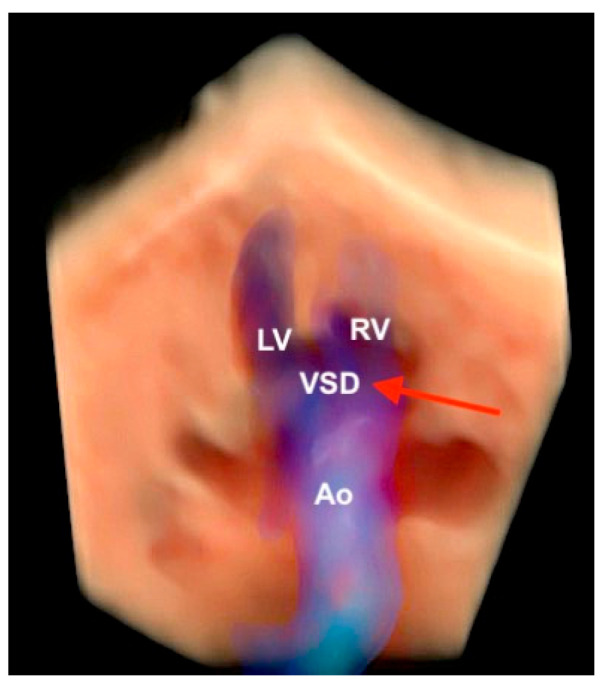
Tetralogy of Fallot using Spatiotemporal Image Correlation with HDlive Flow Silhouette technique in a fetus with trisomy 21 and atrial septal defect. Note the overriding of the aorta (red arrow). Ao: aorta; LV: left ventricle; RV: right ventricle; VSD: ventricular septal defect.

**Figure 22 diagnostics-13-03509-f022:**
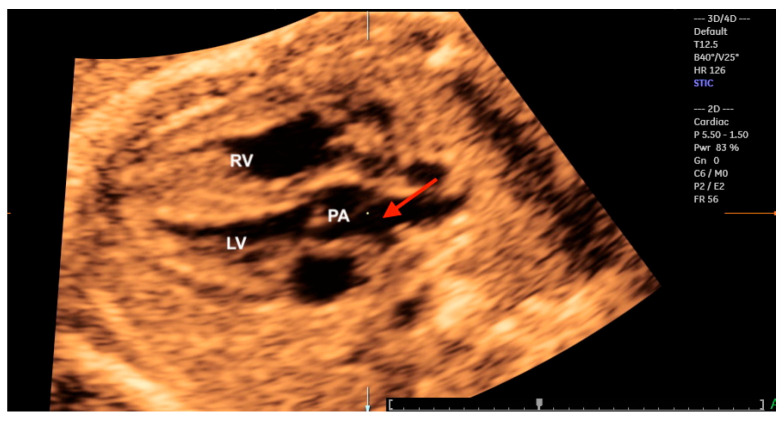
Transposition of the great arteries. Reconstruction of the left ventricular outflow tract using Spatiotemporal Image Correlation in the rendering mode from a 4-chamber view of the fetal heart. Note the bifurcation of the pulmonary artery (red arrow); LV: left ventricle; PA: pulmonary artery; RV: right ventricle.

**Figure 23 diagnostics-13-03509-f023:**
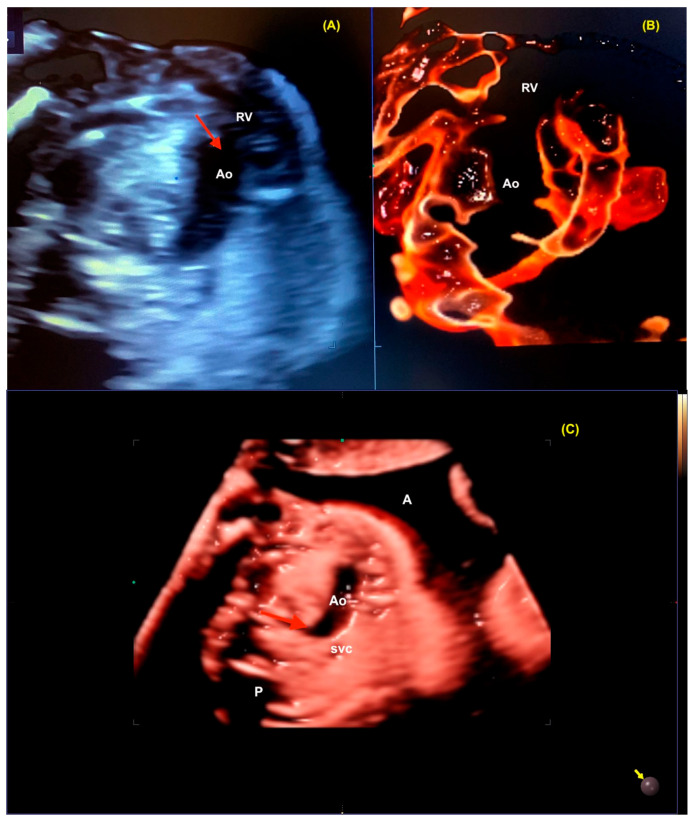
Transposition of the great arteries. (**A**,**B**) Reconstruction of the RV outflow tract from Realistic Vue. Note the aorta (long vessel with reverse curvature) arising from the RV. (**C**) Realistic Vue with Natural Vue and Transparency mode showing only two vessels (aorta and superior vena cava) in the three vessels view (“misnomer 3V”). Ao: aorta (red arrow); RV: right ventricle; svc: superior vena cava; A: anterior; P: posterior.

**Figure 24 diagnostics-13-03509-f024:**
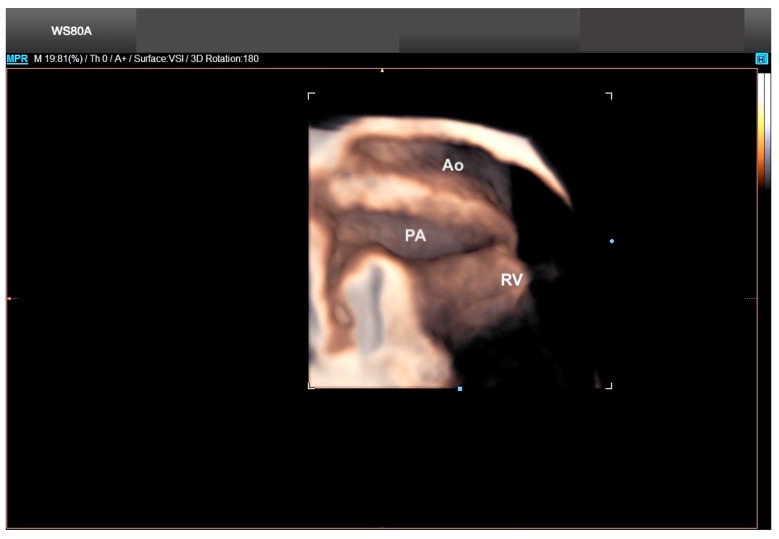
Taussig–Bing anomaly with pulmonary stenosis using Crystal Vue in Volume Shade Imaging mode. Note the anterior aorta (Taussig–Bing), the small pulmonary artery (pulmonary stenosis), and the double outlet tract of the right ventricle. Ao: aorta; PA: pulmonary artery; RV: right ventricle.

**Figure 25 diagnostics-13-03509-f025:**
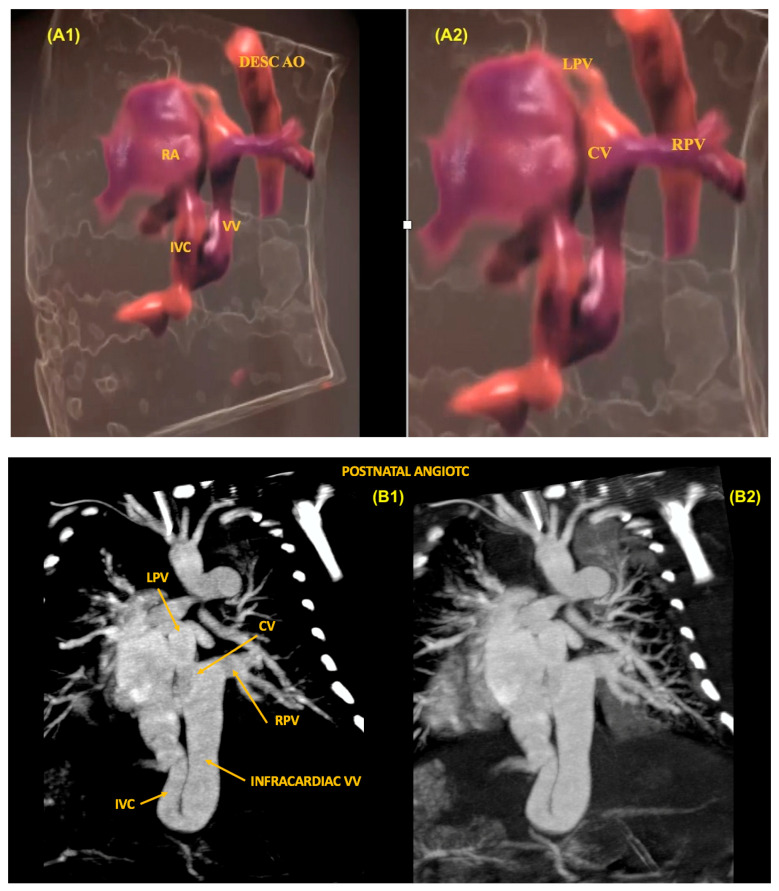
Total Anomalous Pulmonary Vein Return (TAPVR) Type III (infracardiac). (**A1**,**A2**) HDlive Flow technique showing a descending collecting vein with venous confluence (collecting chamber) in a fetus with an infracardiac form of TAPVR. (**B1**,**B2**) Postnatal computed tomography angiography of this case, confirming the findings of prenatal cardiac ultrasound. RA: right atrium; LPV: left pulmonary vein; RPV: right pulmonary vein; CV: collecting venous; Desc Ao: descending aorta.; VV: vertical vein; IVC: inferior vena cava.

**Figure 26 diagnostics-13-03509-f026:**
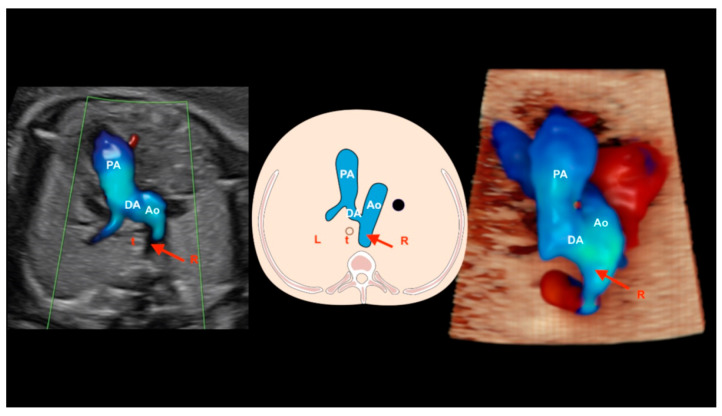
Right aortic arch. Note that the aortic arch is located to the right of the trachea (red arrows). A schematic drawing of the three-vessels and trachea view is shown in the center, an ultrasound image with HDlive Flow is shown on the right, and HDlive Silhouette is shown on the left. PA: pulmonary artery; Ao: aorta; t: trachea; DA: ductus arteriosus; R: right side; L: left side.

## Data Availability

The data presented in this study are available on request from the corresponding author.

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
