# Peer review of "Evolution of Fetal Cardiac Imaging over the Last 20 Years"

_diagnostics, 2023, doi:10.3390/diagnostics13233509_

Round 1

Reviewer 1 Report

Comments and Suggestions for Authors

This is  a quality review of the literature with beautiful illustrations of the technology

Author Response

Reviewer 1.

Open Review

Quality of English Language

( ) I am not qualified to assess the quality of English in this paper
( ) English very difficult to understand/incomprehensible
( ) Extensive editing of English language required
( ) Moderate editing of English language required
( ) Minor editing of English language required
(x) English language fine. No issues detected

Is the work a significant contribution to the field?

Is the work well organized and comprehensively described?

Is the work scientifically sound and not misleading?

Are there appropriate and adequate references to related and previous work?

Is the English used correct and readable?

Comments and Suggestions for Authors

This is  a quality review of the literature with beautiful illustrations of the technology

Submission Date

18 October 2023

Date of this review

18 Oct 2023 18:02:44

Answer: Thank you very much for your considerations.

Reviewer 2 Report

Comments and Suggestions for Authors

The paper is an interesting review of the progress of ultrasound technology for the diagnosis of congenital heart diseases in fetuses. The main aim of fetal diagnosis is to program the appropriate setting for delivery, particularly for complex anatomy which needs procedures immediately after birth or adequate context of observation. Progress in technology improves the quality of images, and the accuracy of diagnosis and will impact future fetal procedures also.

Comments on the Quality of English Language

Need minor editing of language

Author Response

Reviewer 2.

Open Review

Quality of English Language

( ) I am not qualified to assess the quality of English in this paper
( ) English very difficult to understand/incomprehensible
( ) Extensive editing of English language required
( ) Moderate editing of English language required
(x) Minor editing of English language required
( ) English language fine. No issues detected

Is the work a significant contribution to the field?

Is the work well organized and comprehensively described?

Is the work scientifically sound and not misleading?

Are there appropriate and adequate references to related and previous work?

Is the English used correct and readable?

Comments and Suggestions for Authors

The paper is an interesting review of the progress of ultrasound technology for the diagnosis of congenital heart diseases in fetuses. The main aim of fetal diagnosis is to program the appropriate setting for delivery, particularly for complex anatomy which needs procedures immediately after birth or adequate context of observation. Progress in technology improves the quality of images, and the accuracy of diagnosis and will impact future fetal procedures also.

Comments on the Quality of English Language

Need minor editing of language

Submission Date

18 October 2023

Date of this review

05 Nov 2023 12:07:20

Answer: Thank you very much for your considerations. We have checked all text.

Reviewer 3 Report

Comments and Suggestions for Authors

In general, the article is technically well written with a concise expression. The presented topic – the detection of congenital heart disease- is of great interest for obstetricians and health care providers. Indeed, as you have mentioned is the most common birth defect and a leading cause of perinatal mortality related to birth defects.

I consider there are a few things you should improve:

1.     You do not have a paragraph where you explain the “methodology” (Materials and Methods) and criteria you used when you decided what software/technology to present. Did you cover all the existing software? How did you do the research for the article?

2.     Do you consider that the new “advances in 3D assessment of the fetal heart” have improved the detection rate of CHD and please justify your answer with articles / references. Personally, I consider the recent advances in 3D assessment of the fetal heart useful for evaluating the fetal heart especially when trying to characterize and describe a complex CHD, but I don’t think these new techniques present a real advantage for the screening of CHD and for the average obstetrician that performs routine ultrasound screening.

3.     What was the criteria for the selection of the types of CHD you present? You talk about “Anomalous Venous Return”, which is a rare and hard to diagnose congenital heart disease and you do not present any prenatal ultrasound images, but you don’t mention anything about Right aortic arch for example.

4.     You should acknowledge that sometimes 3D images are more difficult to obtain and interpret than standard 2D images, at least for the average examiner. One such example is Figure 23 – A and B.

5.     I suggest reformulating the phrases as you repeat “in addition” : “In addition, through the use of artificial intelligence, the ultrasound machine is able to perform automatic anatomical and functional measurements. In addition, these technologies enable the reconstruction of fetal cardiac structures in realistic images, improving depth perception and resolution of anatomic cardiac details and blood vessels compared to standard two-dimensional (2D) ultrasound.”  

In addition, 3D techniques such as STIC and FINE allow reconstruction (real-time or off-line) of fetal cardiac structures in realistic views based on a basic cardiac ultrasound view, such as the 4C view. In addition, with the use of FINE and artificial intelligence, automatic cardiac ultrasound views and automatic measurements of the anatomical and functional parameters of the fetal heart are provided, highlighting abnormalities.”

In summary, I'd like to congratulate you for your images, and your review addresses a highly significant yet still challenging subject - the detection of CHD. However, I believe there are a few areas that require your attention.

Author Response

Reviewer 3.

Open Review

Quality of English Language

(x) I am not qualified to assess the quality of English in this paper
( ) English very difficult to understand/incomprehensible
( ) Extensive editing of English language required
( ) Moderate editing of English language required
( ) Minor editing of English language required
( ) English language fine. No issues detected

Is the work a significant contribution to the field?

Is the work well organized and comprehensively described?

Is the work scientifically sound and not misleading?

Are there appropriate and adequate references to related and previous work?

Is the English used correct and readable?

Comments and Suggestions for Authors

In general, the article is technically well written with a concise expression. The presented topic – the detection of congenital heart disease- is of great interest for obstetricians and health care providers. Indeed, as you have mentioned is the most common birth defect and a leading cause of perinatal mortality related to birth defects.

I consider there are a few things you should improve:

  1. You do not have a paragraph where you explain the “methodology” (Materials and Methods) and criteria you used when you decided what software/technology to present. Did you cover all the existing software? How did you do the research for the article?

Answer: We do not have a section on methodology because the article is a review. We cover the main existing software, the most widely used in advanced ultrasound   technology.

  1. Do you consider that the new “advances in 3D assessment of the fetal heart” have improved the detection rate of CHD and please justify your answer with articles / references. Personally, I consider the recent advances in 3D assessment of the fetal heart useful for evaluating the fetal heart especially when trying to characterize and describe a complex CHD, but I don’t think these new techniques present a real advantage for the screening of CHD and for the average obstetrician that performs routine ultrasound screening.

Answer: We believe that advanced technologies which include automatic reconstruction of planes and artificial intelligence represent a real advantage for congenital heart disease screening and for the obstetrician performing routine ultrasound screening.

Page 23 Lines 442 to 444 “, especially when the advanced US technologies enable the automatic reconstruction of the Echocardiographic views and artificial intelligence.” 

  1. What was the criteria for the selection of the types of CHD you present? You talk about “Anomalous Venous Return”, which is a rare and hard to diagnose congenital heart disease and you do not present any prenatal ultrasound images, but you don’t mention anything about Right aortic arch for example.

Answer: The criteria for selecting the congenital heart diseases was complexity. In such congenital heart disease, the 3D ultrasound can be useful in detailing the cardiac anatomy. We have added a mention of the right aortic arch and vascular ring (Page 23 Line 432, Figure 26). “Anomalous Venous Return”- Figure 25: this figure should be replaced by the labeled Figure 25 (A) and should be moved closer to Figure 25 (B).

  1. You should acknowledge that sometimes 3D images are more difficult to obtain and interpret than standard 2D images, at least for the average examiner. One such example is Figure 23 – A and B.

Answer: We have added the acknowledgement of this difficulty. Page 20, line 390: "It is important to note that 3D images can sometimes be more difficult to obtain and interpret than standard 2D images for an examiner who is not experienced in advanced US technologies and who is not an expert in cardiac anatomy.

  1. I suggest reformulating the phrases as you repeat “in addition”: “In addition, through the use of artificial intelligence, the ultrasound machine is able to perform automatic anatomical and functional measurements. In addition, these technologies enable the reconstruction of fetal cardiac structures in realistic images, improving depth perception and resolution of anatomic cardiac details and blood vessels compared to standard two-dimensional (2D) ultrasound.”  

In addition, 3D techniques such as STIC and FINE allow reconstruction (real-time or off-line) of fetal cardiac structures in realistic views based on a basic cardiac ultrasound view, such as the 4C view. In addition, with the use of FINE and artificial intelligence, automatic cardiac ultrasound views and automatic measurements of the anatomical and functional parameters of the fetal heart are provided, highlighting abnormalities.”

Answer: We have reworded the sentences in which we repeat "In addition". Page 1- Lines 29 and 30. Page 4 Line 78. Page 10 Line 185.Page 23 Line 438 and 441.

In summary, I'd like to congratulate you for your images, and your review addresses a highly significant yet still challenging subject - the detection of CHD. However, I believe there are a few areas that require your attention.

Round 2

Reviewer 3 Report

Comments and Suggestions for Authors

I still have some doubts about case selection criteria and the way you decided which are "the main existing software" and "the most widely used" software for 3D analysis of the heart, but I respect your decision.

Congratulations for your work!